# Race-ethnicity and COVID-19 Vaccination Beliefs and Intentions: A Cross-Sectional Study among the General Population in the San Francisco Bay Area

**DOI:** 10.3390/vaccines9121406

**Published:** 2021-11-29

**Authors:** Yingjie Weng, Di Lu, Jenna Bollyky, Vivek Jain, Manisha Desai, Christina Lindan, Derek Boothroyd, Timothy Judson, Sarah B. Doernberg, Marisa Holubar, Hannah Sample, Beatrice Huang, Yvonne Maldonado, George W. Rutherford, Kevin Grumbach

**Affiliations:** 1Quantitative Sciences Unit, Department of Medicine, Stanford University, Palo Alto, CA 94304, USA; Di.Lu1@stanford.edu (D.L.); manishad@stanford.edu (M.D.); derekb@stanford.edu (D.B.); 2Division of Primary Care & Population Health, School of Medicine, Stanford University, Stanford, CA 94305, USA; jbollyky@stanford.edu; 3Division of HIV, Infectious Diseases & Global Medicine, San Francisco General Hospital, San Francisco, CA 94110, USA; Vivek.Jain@ucsf.edu; 4Department of Epidemiology and Biostatistics, University of California, San Francisco, CA 94134, USA; Krysia.Lindan@ucsf.edu (C.L.); George.Rutherford@ucsf.edu (G.W.R.); 5Institute for Global Health Sciences, University of California San Francisco, San Francisco, CA 94134, USA; 6Division of Hospital Medicine, Department of Medicine, University of California, San Francisco, CA 94117, USA; Timothy.Judson@ucsf.edu; 7Division of Infectious Diseases, Department of Medicine, University of California, San Francisco CA 94117, USA; Sarah.Doernberg@ucsf.edu; 8Division of Diseases and Geographic Medicine, Stanford University School of Medicine, Stanford, CA 94305, USA; mholubar@stanford.edu; 9Department of Biochemistry and Biophysics, University of California San Francisco, San Francisco CA 94134, USA; hsample@mednet.ucla.edu; 10Department of Family and Community Medicine, University of California, San Francisco, CA 94110, USA; Beatrice.Huang@ucsf.edu (B.H.); Kevin.Grumbach@ucsf.edu (K.G.); 11Division of Pediatric Infectious Diseases, Stanford University School of Medicine , Stanford, CA 94305, USA; bonniem@stanford.edu

**Keywords:** COVID-19 vaccine intention, race–ethnicity, mediators, LASSO

## Abstract

Objective: The study was designed to compare intentions to receive COVID-19 vaccination by race–ethnicity, to identify beliefs that may mediate the association between race–ethnicity and intention to receive the vaccine and to identify the demographic factors and beliefs most strongly predictive of intention to receive a vaccine. Design: Cross-sectional survey conducted from November 2020 to January 2021, nested within a longitudinal cohort study of the prevalence and incidence of SARS-CoV-2 among a general population-based sample of adults in six San Francisco Bay Area counties (called TrackCOVID). Study Cohort: In total, 3161 participants among the 3935 in the TrackCOVID parent cohort responded. Results: Rates of high vaccine willingness were significantly lower among Black (41%), Latinx (55%), Asian (58%), Multi-racial (59%), and Other race (58%) respondents than among White respondents (72%). Black, Latinx, and Asian respondents were significantly more likely than White respondents to endorse lack of trust of government and health agencies as a reason not to get vaccinated. Participants’ motivations and concerns about COVID-19 vaccination only partially explained racial–ethnic differences in vaccination willingness. Concerns about a rushed government vaccine approval process and potential bad reactions to the vaccine were the two most important factors predicting vaccination intention. Conclusions: Vaccine outreach campaigns must ensure that the disproportionate toll of COVID-19 on historically marginalized racial–ethnic communities is not compounded by inequities in vaccination. Efforts must emphasize messages that speak to the motivations and concerns of groups suffering most from health inequities to earn their trust to support informed decision making.

## 1. Introduction

A successful COVID-19 mass vaccination program requires not only sufficient supply of a safe and effective vaccine and well-organized distribution, but also a willingness of people become vaccinated. Vaccination campaigns are more effective and equitable if health officials have a robust understanding of vaccine beliefs and motivations, including variation among sub-populations [1].

National surveys in the United States have found that Black and Latinx individuals have more reservations than their White counterparts about COVID-19 vaccination [2,3,4,5,6]. Lower rates of vaccination among Black and Latinx populations have been attributed to a combination of barriers to access and reduced enthusiasm for vaccination [7,8]. We previously reported our findings from a survey of a general population-based sample and medical center workers in the San Francisco Bay Area, documenting that in both groups, Black, Latinx, Asian, multi-race and other race respondents had significantly lower intention to get a COVID-19 vaccine than White respondents [9]. Here, we expand our analyses of the general population sample, comparing beliefs about COVID-19 vaccination across racial–ethnic groups and the extent to which these beliefs mediated differences across racial–ethnic groups in intention to get vaccinated. We also present data on beliefs and sociodemographic characteristics that were most important in predicting COVID-19 vaccination intention.

## 2. Methods

### 2.1. Study Design and Population

A cross-sectional survey [9] was conducted from late November 2020 to January 2021, nested within a longitudinal cohort study of the prevalence and incidence of SARS-CoV-2 among a probability sample of adults residing within six San Francisco Bay Area counties (called TrackCOVID). Details of study design and implementation are presented elsewhere [10]. Participants in TrackCOVID were enrolled between July and December 2020 and underwent monthly RT-PCR testing of nasopharyngeal swab samples for presence of virus and testing of blood samples for antibodies to SARS-CoV-2. They also completed baseline and monthly surveys on sociodemographics and behaviors.

TrackCOVID used an address-based stratified random sampling strategy to select households eligible for study recruitment. Two strata were considered in the sampling scheme to increase statistical efficiency: estimated COVID-19 cases per census tract determined by modeling, and county. The risk of infection by SARS-CoV-2 for each household was estimated by modeling prevalent cases within census tracts as reported by counties as a function of sociodemographic, occupational, health and poverty characteristics using data from the 2018 American Community Survey and UCSF Health Atlas [11]. One adult from each randomly selected household was eligible for participation.

All participants enrolled in the parent cohort were sent a link to an online survey about COVID-19 vaccination with Research Electronic Data Capture Software (REDCap). Surveys were provided in Spanish, Chinese, Tagalog, and Vietnamese for respondents with limited English proficiency. Those who did not respond to the online electronic instrument were invited to complete the survey in person using a tablet device during a regular testing visit for the parent study, assisted by a research associate if necessary. The survey was administered from 14 December 2020 to 15 January 2021. This timeframe coincided with the announcement of emergency use authorizations of the Pfizer BNT162b2 (11 December 2020) [12] and the Moderna mRNA-1283 (18 December 2020) [13] vaccines.

The TrackCOVID study was designated as a public health surveillance study under 45 CFR 46.102(l) by the Stanford University School of Medicine Administrative Panel on Human Subjects in Medical Research, and the University of California, San Francisco Institutional Review Board.

### 2.2. Survey Instruments

Survey items about vaccination were adapted from the NIH Community Engagement Alliance (CEAL) Against COVID-19 Disparities Draft Common Survey [14] and informed by conceptual models of vaccine hesitancy [15,16]. Questions were asked about perception of risk of becoming infected with SARS-CoV-2, confidence in the vaccine, and motivation to obtain the vaccine, based on the conceptual model of vaccine behavior developed by Brewer et al. [17] building on the Theory of Planned Action model of Fishbein and Ajzen [18].

### 2.3. Primary Outcome

The primary outcome was a participant’s high willingness to receive the vaccine. This binary variable was derived from two survey items. The first item asked, “How likely are you to get an approved COVID-19 vaccine when it becomes available?”, using a 1–7 Likert scale with 1 indicating “not at all likely” and 7 “very likely.” Respondents who scored 3 or greater were asked a second question, “How early would you ideally like to receive the COVID-19 vaccine?”, with response options of “I’d like to be among the earliest,” “I’d like to receive it early, but not in the first round of people,” “I’d like to receive it later in the distribution process,” “or “I’d like to wait at least two months to see what the experience is.” Respondents who selected 3 or greater on the first item and answered “I’d like to be among the earliest” or “I’d like to receive it early…” to the second item were categorized as having “high” vaccination willingness; all others were categorized as having “low” vaccination willingness.

### 2.4. Vaccine Beliefs: Motivators, Concerns and Worries

A set of questions about “motivators” listed reasons that people might want to get a COVID-19 vaccine, asking respondents to rate the importance of each reason on a scale ranging from “Not a reason for me to get the vaccine” to “Most important reason to get the vaccine.” Respondents could pick more than one reason as most important. Responses were then dichotomized to indicate whether that reason was classified as a most important reason. A set of questions about “concerns” about reasons to not get the vaccine, with response options ranging from “Not a reason” to “Most important reason to not get the vaccine.” Responses were dichotomized to indicate whether the concern was a most or moderately important consideration in not getting the vaccine. Two additional items measured degree of worry that a COVID-19 vaccine “Might not stop you from getting COVID” and “Might give you COVID and make you sick,” with responses dichotomized (very or somewhat worried vs. neutral or not at all). (Questions and possible responses are listed in Appendix B Table A1). We considered the above beliefs about COVID vaccines as potential mediators for the association between race–ethnicity and vaccine willingness.

### 2.5. Sociodemographic Variables

Participants completed a survey at their baseline enrolment visit in the parent cohort study that included questions about socio-demographics. Participants self-identified their race–ethnicity using Office of Management and Budget (OMB) categories, with one item asking about Hispanic/Latinx identity and one item asking about race with response options of White, Black or African American, American Indian or Alaska Native, Asian, Native Hawaiian or Other Pacific Islander, and Other race. Participants could select more than one race. The ethnicity and race items were then combined to create a single variable with mutually exclusive categories (White, Black, Latinx, Asian, Mixed race–ethnicity, and Other race–ethnicity). Because of the small number of American Indian and Native Hawaiian respondents, these respondents were included in the Other race–ethnicity category. The few respondents who identified as both Hispanic/Latinx and Black were categorized as Black. Other sociodemographic variables included age, gender, occupation, employment status, and highest level of education attained.

### 2.6. Statistical Methods

#### 2.6.1. Descriptive Analysis

Frequencies and means of the social-demographic variables were described for our study cohort. Logistic regression was employed to characterize the association between race–ethnicity and each vaccine belief (motivator, concern, worries), with calculation of adjusted odds ratios (aOR) and 95% confidence intervals (CI). Regression models adjusted for age, gender and education.

#### 2.6.2. Mediation Analysis

Poisson regression was used to evaluate whether COVID vaccine beliefs mediated the association between race–ethnicity and vaccine willingness. Adjusted prevalence ratios (aPR) and the corresponding 95% CI of vaccine willingness were estimated from the models. Specifically, a step-wise Poisson model was performed that first only included race/ethnicity and other demographic characteristics (age, gender, education) as predictors of high vaccine willingness. Then, a second model was developed in which motivators, concerns and worries were also included as potential predictors. aPRs for race–ethnicity from the two models were compared to identify if participants’ beliefs about COVID-19 vaccines mediated the observed race–ethnicity disparity. Complete case analysis was applied for this analysis.

#### 2.6.3. Least Absolute Shrinkage and Selection Operator (LASSO) Model

A statistical learning method was employed using LASSO [19] regression to identify which beliefs and sociodemographic features were most important in predicting high vaccine willingness. The sample and corresponding data were randomly split in a 1:1 ratio into two datasets—one was used for “training” (or developing the model), and one was used for “testing” the model. LASSO approaches were then applied with 5-fold cross-validations in the training sample; the testing sample was used to evaluate the performance of the model. Area under the curve (AUC) and C-index were calculated as the performance index for both training and testing sets. A range of values for the regularization parameter (**λ**) was evaluated in the training sample and the optimal **λ** that minimized the change of the performance index was chosen to build the final model. Categories within variables were grouped together using grouped LASSO techniques such that each of the input variables would be either chosen or not chosen as a whole instead of by each individual subgroup within that variable [19]. For example, the model would either select or not select race–ethnicity using this approach, rather than having an indicator for “Black” selected but an indicator of “Latinx” not selected as a predictor of high willingness to receive the vaccine. AUC and its corresponding 95% CI calculated through bootstrapping were reported from the final validation model using the testing dataset. Variable importance was estimated using the absolute value of the β coefficients from the LASSO model in the training set. The variable importance for each of the selected predictors was ranked in descending order and presented using a funnel plot. Multiple imputations were performed prior to LASSO regressions to handle variables with missing data.

#### 2.6.4. Sensitivity Analysis

Sensitivity analyses using alternative measures of willingness as the outcome variable were performed. The alternative measure of high willingness was a single item asking respondents to rate their degree of agreement with the statement, “I plan to wait and see how it goes with people who first get the vaccine before getting a vaccine myself.” Respondents who strongly disagreed or disagreed were categorized as having high vaccination willingness. We also created a high hesitancy measure, based on reporting a low likelihood of getting vaccinated.

All statistical analyses were performed using R Statistical Programming Languages, version 4.0.3 and SAS, version 9.4 (SAS Institute, Cary, NC, USA).

## 3. Results

All 3935 participants in the Track COVID study were sent the vaccine survey, of whom 3161 (80.3%) responded. Those who responded were more likely to be older, White, and highly educated compared to non-responders (Appendix B Table A2). Among survey respondents, 3 (0.1%) did not report race–ethnicity and were excluded from the analysis. The total sample used for regression analysis was 3158.

Table 1 shows the demographic characteristics of the study sample. The mean age of survey respondents was 47.6 years old (SD: 14.8); 53.8% were female, 61.0% were White, 3.7% were Black, 9.9% were Hispanic/Latinx, and 18.2% were Asian. Participants were highly educated, with 87.5% having a at least a college degree; 8.2% were employed in the healthcare sector.

### 3.1. Vaccine Willingness

Overall, Black, Latinx, Asian, Multi-racial, and Other race respondents were significantly less likely than White respondents to report high willingness to be vaccinated (Table 2).

### 3.2. Association between Race–Ethnicity and Vaccine Beliefs

Despite differences in willingness to get vaccinated, reasons to get vaccinated were similar across racial–ethnic groups (Table 3). Latinx and Asian respondents were, however, significantly more likely than White respondents to report “my doctor’s recommendation” as the most important reason.

Black, Latinx, and Asian respondents were more likely than White respondents to endorse reasons not to get vaccinated (Table 3). Compared with White, about 20% more of the Black, Latinx, and Asian respondents endorsed “bad reaction to the vaccine” and “government rushing the approval process” as major reasons not to get vaccinated. About one quarter of Black respondents endorsed lack of trust in the companies making COVID-19 vaccines, and lack of belief that a vaccine would be effective in preventing infection as major reasons not to get vaccinated. Black, Latinx, Asian, multi-racial, and other race respondents were much more likely than White respondents to report concerns about the vaccine giving them COVID-19.

### 3.3. Beliefs as Mediators for the Association between Race–Ethnicity and Vaccine Willingness

Vaccine beliefs that were found to be significantly associated with high willingness to be vaccinated included a desire to protect one’s family and oneself from COVID-19, and the belief that life will not return back to normal until most people get the vaccine (Table 4). The factors negatively associated with high willingness to get vaccinated included concerns about the government rushing the approval process, obtaining a bad reaction from the vaccine, and getting COVID-19 from the vaccine. Comparing the aPR for racial–ethnic groups for the first regression model without potential mediators, and the second regression model with all variables, indicates whether the motivators and concerns explain a portion of the association between race–ethnicity and high willingness to be vaccinated. The aPRs for each of the racial–ethnic groups are closer to 1 (or no association) in the second model, suggesting that the vaccine beliefs included in the survey explain at least a small portion of lower vaccine willingness for Black, Latinx and Asian respondents relative to White.

### 3.4. Top Sociodemographic and Vaccine Belief Predictors of Vaccination Willingness

The LASSO model identified the following as the most important factors in explaining COVID-19 vaccination intention (in rank order): concern about government rushing the vaccine process, concern about having a bad reaction to the vaccine, concern about getting COVID-19 from the vaccine, motivation to protect one’s self and family from COVID-19, belief that life will only go back to normal when most people get the vaccine, and willingness to get the vaccine if recommended by one’s doctor (Figure 1). The AUC of the LASSO model was 0.762 (95% CI: 0.737–0.788).

### 3.5. Sensitivity Analyses

Patterns of associations between race–ethnicity, beliefs, and vaccination willingness were essentially the same when we used the alternative measure of high willingness, and the measure of low intention, as our outcome variables. For example, although the alternative measure of high willingness gave a lower point estimate of willingness than the measure included in the results reported above, both measures had similar associations with race–ethnicity and vaccination reasons. Top predictors selected by LASSO were slightly different for the alternative measure of high willingness. The most important factors included (in rank order): concern about having a bad reaction to the vaccine and concern about government rushing the vaccine process (Appendix B Figure A1).

## 4. Discussion

The findings of this study are consistent with previous research that has found lower intention to receive a COVID-19 vaccine among Black and Latinx populations [2,3,4,5,6,7,8,9,20,21,22,23,24,25,26,27,28]. Many of those studies were conducted early in the vaccine development stage, months before emergency use authorization in the US of the first COVID-19 vaccines. Our survey period straddled the dates of emergency use authorization of the Moderna and Pfizer-BioNTech vaccines—a time when decision making about vaccination was close to the start of actual vaccine availability for health care workers, but not for the majority of the US public. The study sample was designed to include diverse racial–ethnic groups from the general population of the six bay area counties, allowing us to compare intentions and analyze mediating factors within the largest racial–ethnic groups including White, Black, Asian and Latinx. Asian, multi-racial, and other race–ethnicity respondents were found to be more similar to Black and Latinx participants than to White respondents regarding their intentions and beliefs about COVID-19 vaccination. This is one of the few published studies [3,20,25,28] to test the independent contribution of a broad range of beliefs as predictors of intentions to seek COVID-19 vaccination. In addition, the study applied both multivariate regression techniques and statistical learning methods to identify potential mediators of racial–ethnic differences as well as to identify predictors of intention to obtain COVID-19 vaccination.

Motivations to seek vaccination were comparable among respondents across all racial–ethnic groups in our study. Similar to findings from other studies, the most important reasons to get vaccinated were a desire not only to protect oneself, but also one’s family and community [29]. Fear of COVID-19 infection and the potential economic, physical and mental health consequences associated with restrictive measures, such as quarantine, lockdown and stay-at-home orders imposed due to continued transmission, could also prompt people to get vaccinated, although our study did not evaluate these factors [30,31,32,33,34,35,36,37]. A preponderance of study participants in all racial–ethnic groups endorsed the gravity of the pandemic, with few agreeing with the statement “The COVID-19 outbreak is not as serious as some people say it is” as a reason to not get vaccinated. The most prevalent concerns about COVID-19 vaccination among respondents to our survey were also similar to ones reported in other studies [21,24,25,26,28], with concerns about efficacy and safety being most commonly endorsed. Whereas responses to reasons to get vaccinated were comparable across racial–ethnic groups in our study, Black, Latinx, and Asian respondents were much more likely than White respondents to endorse reasons to not get vaccinated, and in particular to endorse lack of trust as a reason to not get vaccinated.

Differences in motivations and concerns about COVID-19 vaccination across racial–ethnic groups only partly explained the observed racial–ethnic disparities in vaccine intentions, suggesting that there are additional unmeasured motivations and concerns across groups that mediate differences in vaccination intentions. Among the items measured, concern about a rushed vaccine approval process emerged as the single strongest predictor of lower willingness to get vaccinated, in both the regression and LASSO models. Trust in government has been found to be a factor associated with COVID-19 vaccine intentions among surveys from other countries [28,29].

COVID-19 vaccines were rapidly developed, tested, and approved for emergency use authorization under the pressure of the pandemic’s devastating global health impact. Extensive resources were established to facilitate the development of the vaccines from multiple pipelines and platforms and to ensure the scientific rigor of the development and clinical trial process [38,39,40,41]. At the time this survey was conducted, phase III clinical trials had provided strong evidence of the safety and efficacy of the initial COVID-19 vaccines, justifying emergency use authorization [42,43,44,45,46]. However, findings from our study suggest that some individuals were taking a “wait and see” approach, wanting more information from post-market surveillance to reassure them about effectiveness and safety. Phase IV observational studies in the US and other nations have in fact borne out the excellent effectiveness of COVID-19 vaccines in preventing severe disease, along with the very low incidence of serious adverse reaction [37,38,39,40,41,42,43,44,45,46,47,48,49]. Unfortunately, ongoing challenges with vaccine uptake have made it clear that the problem is not only insufficient information, but pervasive misinformation [50,51,52,53,54,55].

Study limitations include survey administration relatively early in the roll-out of the Pfizer and Moderna vaccines in the US. As such, our data do not reflect shifts in viewpoints, increasing confidence, or other changes in opinions in subsequent months as the US vaccination campaign expanded. The survey sample was drawn from people sufficiently concerned about their risk of COVID-19 and who trusted the study process enough to volunteer for a survey that required repeated COVID-19 testing. Self-selection and non-response may bias results due to persons being more willing to be vaccinated, also being more willing to enroll in the study. However, self-selection and non-response are less likely to introduce bias when testing associations of variables within the study cohort, such as associations between race–ethnicity and vaccine intentions or between beliefs and intentions. It is striking that even within groups of individuals motivated to participate in a longitudinal COVID-19 study, large racial–ethnic differences exist in COVID-19 vaccination intentions and reasons not to get vaccinated.

The parent cohort was designed to obtain a representative population-based sample of adults from a large urban region of California, encompassing six counties with a total adult population of 5,321,907. At the time of the vaccine survey and analysis, neither the parent cohort nor the vaccine survey sample had been weighted to census level demographic characteristics. The intention of TrackCOVID was to inform public health surveillance in the region, and not to be generalizable to the US overall. Nevertheless, the parent study is one of the few population-based studies of SARS-CoV-2 infection that had been performed in the US. Given the high response rate of those in the vaccine survey, our results can reasonably represent the study region, if not broadly generalizable to other areas of the US. Additional limitations include the lack of inclusion of other factors that may influence intention to seek vaccination such as concerns about vaccine access. Finally, our primary outcome was self-reported vaccine intention, which may or may not reflect actual vaccine uptake once vaccines became more available and eligibility was expanded.

The study has important implications for COVID-19 vaccination strategies. Findings highlight that special effort is required to reach historically marginalized populations to support informed decision-making about vaccination. These campaigns must forthrightly acknowledge the history of racism in biomedical research and healthcare delivery that has harmed race–ethnicity minority groups, degraded the trustworthiness of health and medical science institutions, and continues to undermine confidence in vaccines [56,57]. Although the study suggests that educational outreach must address common concerns about vaccine risk and efficacy, it also points to positive messages that may resonate. Altruism was a strong motivator for vaccination, especially the desire to protect one’s family. Vaccination decision-making is a deliberative and dynamic process. This may be particularly salient for interpreting the powerful effect on intentions of concern about a rushed vaccine approval process. This concern may be mitigated over time as more people are vaccinated and evidence accumulates about vaccine efficacy and safety. Finally, it is important to emphasize that addressing motivations and concerns must not distract from the importance of ensuring more equitable access to vaccination [8,58]. Many individuals in all racial–ethnic groups have high willingness to be vaccinated but face barriers to obtaining vaccination [7,59].

In summary, vaccine outreach campaigns must ensure that the disproportionate toll of COVID-19 on communities of color is not compounded by inequities in vaccination. Efforts must emphasize messages that speak to the motivations and concerns of groups suffering most from health inequities and ensure equitable access to vaccinations.

## 5. Conclusions

Vaccine outreach campaigns must ensure that the disproportionate toll of COVID-19 on historically marginalized racial–ethnic communities is not compounded by inequities in vaccination. Efforts must emphasize messages that speak to the motivations and concerns of groups suffering most from health inequities to earn their trust to support informed decision making.

## Figures and Tables

**Figure 1 vaccines-09-01406-f001:**
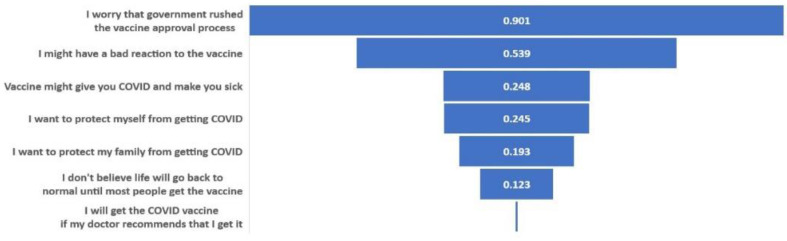
Relative Importance of Top Predictors for Vaccine Willingness, based on LASSO Regression Results, Variable importance shown here were defined as absolute values of the β coefficients for the selected predictors in LASSO model.

**Table 1 vaccines-09-01406-t001:** Characteristics of survey respondents.

Demographic Characteristics	Survey Respondents (N = 3161)
	*n* (%)
**Age, years**	
18–39	885 (28.0)
40–64	1543 (48.5)
≥65	742 (23.5)
*Age, years (mean (SD))*	51.1 (15.8)
**Gender**	
Female	1702 (53.8)
Male	1431 (45.3)
Other ^1^	27 (0.9)
Unknown	1 (0)
**Race–ethnicity Group**	
White	1928 (61.0)
Black	116 (3.7)
Hispanic/Latinx	312 (9.9)
Asian	575 (18.2)
Multiple races	154 (4.9)
Other	73 (2.3)
Unknown	3 (0.1)
**Education**	
Less than college	340 (10.8)
College ^2^	1506 (47.6)
Higher than college	1261 (39.9)
Unknown	54 (1.7)
**Occupation**	
Employed in health sector	258 (8.2)
Not employed in health sector	2903 (91.8)

^1^ Self-reported gender group: other gender group included trans male, trans female, or genderqueer/gender non-binary. ^2^ Includes associate degree (community college) and bachelor’s degree.

**Table 2 vaccines-09-01406-t002:** High COVID-19 vaccination willingness, by race–ethnicity.

Race-Ethnicity Groups	Number of Respondents	Percentage Reporting High Vaccine Willingness (95% CI)
All Respondents	3161	66% (64–67%)
White	1928	72% (70–74%)
Black/African American	116	41% (32–50%)
Latinx/Hispanic	312	55% (49–60%)
Asian	575	58% (54–62%)
Other Race	73	58% (46–69%)
Multiple Races	154	59% (51–67%)

**Table 3 vaccines-09-01406-t003:** Comparison of vaccine motivators and concerns, by race–ethnicity (reference group: White).

Outcomes	White (*n* = 1928)	Black (*n* = 116)		Asian (*n* = 575)		Hispanic/Latinx (*n* = 312)		Multiple Races (*n* = 154)		Other Races (*n* = 73)	
	*n* (%)	*n* (%)	aOR [95% CI]	*n* (%)	aOR [95% CI]	*n* (%)	aOR [95% CI]	*n* (%)	aOR [95% CI]	*n* (%)	aOR [95% CI]
**Reasons to get vaccinated (motivators)**, % indicating most important reason											
a. I want to protect myself from getting COVID	1422 (74.2)	86 (74.1)	1.01 (0.65, 1.57)	418 (73.1)	1.22 (0.98, 1.53)	208 (67.3)	0.93 (0.71, 1.23)	99 (64.3)	0.86 (0.60, 1.24)	50 (68.5)	0.90 (0.53, 1.53)
b. I want to protect my family from getting COVID	1522 (79.6)	89 (77.4)	0.81 (0.52, 1.28)	465 (81.0)	1.12 (0.87, 1.42)	256 (83.1)	1.20 (0.86, 1.67)	127 (82.5)	1.19 (0.77, 1.86)	55 (75.3)	0.79 (0.45, 1.39)
c. I want to protect my community from getting COVID	1317 (69.1)	73 (64.0)	0.75 (0.50, 1.12)	367 (64.2)	0.86 (0.70, 1.06)	211 (68.3)	0.96 (0.73, 1.26)	103 (66.9)	0.90 (0.63, 1.29)	47 (64.4)	0.81 (0.49, 1.34)
d. I don’t believe life will go back to normal until most people get the vaccine	1221 (64.0)	68 (59.1)	0.81 (0.55, 1.20)	356 (62.2)	1.05 (0.86, 1.28)	188 (61.0)	0.95 (0.73, 1.24)	87 (56.5)	0.83 (0.59, 1.16)	43 (58.9)	0.93 (0.56, 1.52)
e. I will get the COVID vaccine if my doctor recommends that I get it	617 (32.4)	47 (41.2)	1.42 (0.95, 2.12)	221 (38.6)	1.60 (1.31, 1.97)	109 (35.9)	1.26 (0.96, 1.66)	52 (33.8)	1.30 (0.90, 1.86)	22 (30.1)	1.01 (0.60, 1.69)
f. My employer or school will require or expect me to get a vaccine	300 (15.9)	17 (15.3)	0.89 (0.52, 1.52)	130 (23.1)	1.57 (1.23, 1.99)	67 (22.0)	1.33 (0.97, 1.82)	32 (21.2)	1.26 (0.82, 1.93)	15 (20.8)	1.28 (0.70, 2.35)
g. My family will want me to get a vaccine	653 (34.5)	36 (32.1)	0.81 (0.53, 1.25)	208 (36.6)	1.38 (1.12, 1.70)	106 (34.9)	1.16 (0.88, 1.53)	49 (32.2)	1.13 (0.78, 1.64)	28 (38.4)	1.20 (0.72, 1.99)
**Reasons to not get vaccinated (concerns)**, % indicating most important or moderately important reason											
a. The vaccine may not stop me from getting COVID	224 (11.8)	31 (27.7)	2.62 (1.67, 4.10)	104 (18.3)	1.68 (1.29, 2.19)	74 (24.3)	2.16 (1.58, 2.96)	26 (17.2)	1.48 (0.93, 2.35)	12 (16.9)	1.38 (0.71, 2.69)
b. I might have a bad reaction to the vaccine	466 (24.5)	54 (49.1)	2.56 (1.72, 3.81)	250 (44.0)	2.39 (1.95, 2.93)	132 (43.1)	2.05 (1.57, 2.66)	54 (35.3)	1.49 (1.04, 2.14)	31 (42.5)	2.07 (1.27, 3.38)
c. I do not get vaccines in general	81 (4.3)	18 (16.2)	3.96 (2.26, 6.94)	49 (8.7)	2.19 (1.50, 3.19)	34 (11.2)	2.56 (1.65, 3.99)	12 (7.8)	1.69 (0.87, 3.28)	4 (5.6)	1.30 (0.46, 3.67)
d. I do not think I will get COVID, even without getting a vaccine	39 (2.1)	16 (14.5)	8.05 (4.27, 15.2)	38 (6.7)	3.58 (2.23, 5.75)	21 (6.9)	3.45 (1.94, 6.14)	4 (2.6)	0.96 (0.29, 3.20)	4 (5.6)	2.15 (0.64, 7.19)
e. The COVID-19 outbreak is not as serious as some people say it is	37 (1.9)	14 (12.5)	6.06 (3.08, 11.9)	31 (5.5)	2.74 (1.66, 4.53)	23 (7.6)	3.82 (2.17, 6.72)	6 (3.9)	1.70 (0.65, 4.47)	2 (2.8)	1.41 (0.33, 6.01)
f. I do not trust the companies making COVID-19 vaccines	142 (7.5)	28 (24.8)	3.60 (2.24, 5.78)	72 (12.8)	1.70 (1.25, 2.32)	54 (17.7)	2.21 (1.54, 3.15)	26 (17.1)	1.96 (1.20, 3.19)	10 (13.9)	1.50 (0.70, 3.21)
g. It is better to become immune to a disease by getting sick than by getting a shot	42 (2.2)	10 (8.8)	4.11 (1.99, 8.50)	32 (5.7)	2.86 (1.76, 4.64)	25 (8.3)	3.86 (2.25, 6.62)	10 (6.6)	3.00 (1.41, 6.40)	4 (5.6)	2.67 (0.92, 7.69)
h. I worry that government rushed the vaccine approval process	344 (18.2)	43 (37.7)	2.39 (1.59, 3.61)	190 (33.6)	2.01 (1.61, 2.50)	108 (35.4)	1.87 (1.42, 2.47)	49 (32.2)	1.59 (1.09, 2.32)	20 (28.6)	1.35 (0.76, 2.39)
**Worries about vaccine (worries)**, % strongly agreeing or agreeing											
Might not stop you from getting COVID	502 (26.2)	46 (39.7)	1.73 (1.16, 2.57)	191 (33.2)	1.42 (1.15, 1.75)	121 (39.3)	1.74 (1.34, 2.27)	48 (31.2)	1.21 (0.83, 1.74)	26 (35.6)	1.36 (0.81, 2.27)
Might give you COVID and make you sick	247 (12.9)	43 (38.1)	3.69 (2.44, 5.59)	135 (23.6)	2.27 (1.78, 2.90)	104 (33.8)	3.01 (2.26, 4.03)	38 (24.8)	2.01 (1.33, 3.04)	19 (26.4)	2.45 (1.41, 4.27)

aOR, adjusted odds ratio, from logistic regression models including covariates for age, gender and education. Reference group is White.

**Table 4 vaccines-09-01406-t004:** Results of stepwise Poisson model of demographics, and beliefs as predictors of high vaccine willingness.

Predictors	Outcome: High Willingness to Get Vaccinated	
	Demographic Only Model (aPR [95% CI])	Full Model (aPR (95% CI))
**Race–ethnicity** (ref: White)		
Asian	0.82 (0.73, 0.93)	0.87 (0.77, 1.00)
Black	0.62 (0.46, 0.84)	0.72 (0.52, 1.00)
Hispanic/Latinx	0.83 (0.68, 1.01)	0.93 (0.76, 1.14)
Multiple races	0.80 (0.67, 0.96)	0.86 (0.72, 1.04)
Other	0.85 (0.63, 1.16)	0.91 (0.65, 1.26)
**Belief Mediators**		
** Motivators **		
I want to protect myself from getting COVID		1.22 (1.08, 1.38)
I want to protect my family from getting COVID		1.23 (1.07, 1.43)
I want to protect community from getting COVID		0.97 (0.86, 1.09)
I don’t believe life will go back to normal until most people get the vaccine		1.16 (1.04, 1.29)
I will get the COVID vaccine if my doctor recommends that I get it		1.03 (0.92, 1.15)
My employer or school will require or expect me to get a vaccine		1.09 (0.95, 1.25)
My family will want me to get a vaccine		1.02 (0.90, 1.14)
** Concerns **		
The vaccine may not stop me from getting COVID		0.89 (0.74, 1.07)
I might have a bad reaction to the vaccine		0.76 (0.67, 0.87)
I do not get vaccines in general		0.92 (0.68, 1.22)
I do not think I will get COVID, even without getting a vaccine		0.97 (0.67, 1.40)
The COVID-19 outbreak is not as serious as some people say it is		1.06 (0.74, 1.53)
I do not trust the companies making COVID-19 vaccines		0.89 (0.69, 1.14)
It is better to become immune to a disease by getting sick than by getting a shot		1.13 (0.77, 1.64)
I worry that government rushed the vaccine approval process		0.63 (0.54, 0.75)
** Worries **		
COVID-19 vaccine might not stop you from getting COVID		1.03 (0.92, 1.15)
Might give you COVID and make you sick		0.84 (0.71, 0.99)

aPR, adjusted prevalence ratio. Both models adjusted for age, gender, and education.

## Data Availability

All study materials are available for request upon approval of the corresponding author.

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
