# Peer review of "Race-ethnicity and COVID-19 Vaccination Beliefs and Intentions: A Cross-Sectional Study among the General Population in the San Francisco Bay Area"

_vaccines, 2021, doi:10.3390/vaccines9121406_

Round 1

Reviewer 1 Report

Comparing this kind of report, the authors' survey was conducted in a very limited area. 

Author Response

Response:

We appreciate the reviewer’s feedback. Our survey was conducted within the 6 county region of the San Francisco Bay Area. We acknowledged in the Discussion section of our original submission that this is a study limitation. With that caveat mind, we believe our findings provide helpful information to public health agencies and other stakeholders on the topic vaccine willingness and reluctance.

Reviewer 2 Report

I read with great interest the article entitled ‘Race-ethnicity and COVID-19 Vaccination Beliefs and Intentions: A Cross-sectional Study Among the General Population 3 in the San Francisco Bay Area’.

  • The article is very interesting and helps to better understand what are the reasons for reluctance to carry out the vaccine. I don't have many observations to make on introduction, methods and results. As regards the discussion, I ask the authors to introduce a possible theme: access to information. Did the sample examined have the opportunity to correctly inform themselves about COVID-19? In the authors' opinion, can the lack of awareness of the lethality of the disease and the absence of fear partly explain the differences? In other contexts, the fear of COVID-19 has also had important repercussions on hospital admissions [Fear of the COVID-19 and medical liability. Insights from a series of 130 consecutives medico-legal claims evaluated in a single institution during SARS-CoV-2-related pandemic." (2021).].
  • Elsewhere, in addition to the fear of COVID-19, one of the reasons that prompted the population to get vaccinated was the fear both from a health and economic point of view of new stringent restrictive measures. ["The“ quarantine dry eye ”: the lockdown for coronavirus disease 2019 and its implications for ocular surface health." Risk management and healthcare policy 14 (2021): 1629.]. Is there a difference in work background between the "white" population and the "Latino" or "black" population?The need to work may have influenced their choices?
  • I ask the authors to discuss these two issues very concisely in the discussion and to expand the number of references.

Overall I find the paper very interesting and I congratulate the authors for the analysis that takes into account minorities and tries to understand their motivations.

Author Response

Response:

We appreciate the reviewer’s insightful suggestions.

Our survey instrument did not include items directly measuring access to information. However, the instrument did include items that have a bearing on the reviewer’s question about participants’ “awareness of the lethality of the disease.” Results included in Table 3 strongly suggest that most participants had a good understanding of COVID-19 as a fearsome pandemic. We added the following sentence in the Discussion section that emphasizes one of the key items shown in Table 3:

“A preponderance of study participants in all racial-ethnic groups endorsed the gravity of the pandemic, with very few agreeing with the statement “The COVID-19 outbreak is not as serious as some people say it is” as a reason to not get vaccinated.” (line 275)

We thank the reviewer for raising the issue of information and misinformation, and expanded our discussion of those issues in the Discussion section:

“COVID-19 vaccines were rapidly developed, tested, and approved for emergency use authorization under the pressure of the pandemic’s devastating global health impact. Extensive resources were put in place to facilitate the development of the vaccines from multiple pipelines and platforms and to ensure the scientific rigor of the development and clinical trial process.30-33 At the time this survey was conducted, phase III clinical trials had provided strong evidence of the safety and efficiency of the initial COVID-19 vaccines, justifying emergency use authorization.34-38. However, findings from our study suggest that some individuals were taking a “wait and see” approach, wanting more information from post-market surveillance to reassure them about efficacy and safety.  Phase V observational studies in the US and other nations have in fact borne out the excellent efficacy of COVID-19 vaccines in preventing severe disease, along with the very low incidence of serious adverse reactions39-41.  Unfortunately, ongoing challenges with vaccine uptake have made it clear that the problem is not only insufficient information, but pervasive misinformation. 42-47” (line 293)

In terms of the reviewer’s question about occupational status and related socioeconomic factors potentially influencing differences among racial-ethnic groups, we did include level of education in all our multivariate models, thereby controlling for possible confounding by this one powerful measure of SES. To explore this further, we looked the distribution of work categories by race-ethnicity group. In TrackCOVID cohort, 19% of the black population, 27% of the Latino population were essential workers (vs 18% in white population).

We expanded the literature review as suggested and incorporated many additional citations [Please see the attached file].

Reviewer 3 Report

The manuscript presents a research that compares the intention to receive COVID-19 vaccination by race-ethnicity, to identifies beliefs that may mediate the association between race-ethnicity and intention to receive the vaccine, and to identify the demographic factors and beliefs most strongly predictive of intention to receive a vaccine

The results showed statistical evidence of resistence from an ethnical group and the main reason was lack of trust on goverment rushed approval process.

I find the topic interesting and being worth of investigation and the document is well strucutred, organized, fluidly written, the methodology followed is clearly explained, the sample is very representative, the results are clearly presented and support the conclusions.

Although I propose the following suggestions / considerations:
- I strongly suggest authors from refraining using personal pronouns such as "we" and "our" throughout the text and I encourage them to write it in an impersonal form of writing.

Author Response

Response:

We appreciate the reviewer’s suggestions and have removed almost all first person plural pronouns and revised using impersonal language.

Reviewer 4 Report

The authors propose a study on a very relevant issue in relation to the control of the pademic generated by covid-19. In this regard, the study is interesting, the methodology used is adequate and consistent with the purpose of the study. The results are well shown and support the discussion. However, I have the following comments (minor comments).

I. Minor comments:
1. Improve the writing of the study objective.
2. In the introduction I suggest including a brief paragraph regarding the vaccines that have been developed and used to control the pandemic. Emphasizing that vaccines have proven to be safe and efficient.
3. In the discussion it is necessary to include a paragraph (discuss) regarding the efficiency and safety of vaccines in those countries where a significant percentage of the population has been vaccinated. This is an important aspect of reporting regarding vaccine safety. Vaccines are safe and efficient, but unfortunately, anti-vaccine groups rely on lies and unscientific theories to question vaccines.

Author Response

Response:

We appreciate the insightful suggestions bought up by the reviewer and strongly agree with the points made by the reviewer. We incorporated the following revisions of our manuscript.

  1. We revised our study objective as follows.

[Line 21] “The study was designed to compare intention to receive COVID-19 vaccination by race-ethnicity, to identify beliefs that may mediate the association between race-ethnicity and intention to receive the vaccine, and to identify the demographic factors and beliefs most strongly predictive of intention to receive a vaccine.”

  1. We included a paragraph presenting evidence on the safety and efficiency of covid-19 vaccine in the discussion section

[Line 293] ‘COVID-19 vaccines were rapidly developed, tested, and approved for emergency use authorization under the pressure of the pandemic’s devastating global health impact. Extensive resources were put in place to facilitate the development of the vaccines from multiple pipelines and platforms and to ensure the scientific rigor of the development and clinical trial process.30-33 At the time this survey was conducted, phase III clinical trials had provided strong evidence of the safety and efficiency of the initial COVID-19 vaccines, justifying emergency use authorization.34-38. However, findings from our study suggest that some individuals were taking a “wait and see” approach, wanting more information from post-market surveillance to reassure them about efficacy and safety.  Phase V observational studies in the US and other nations have in fact borne out the excellent efficacy of COVID-19 vaccines in preventing severe disease, along with the very low incidence of serious adverse reactions39-41

Round 2

Reviewer 1 Report

I do not agree with their very limited study design, even if they describe that in discussion. 

Author Response

Response:

We appreciate the reviewer’s feedback. However, we are unable to redesign or reimplement the study. All studies have limitations, and we have tried to clearly identify them in the Discussion section, and have expanded on issues of generalizability (Lines 324-337). Similar studies using regional or single state survey samples to study vaccine acceptability have been reported in the literature, and have provided insightful data for public health decision making at the regional level (see references listed below).

Even though the parent study and the vaccine survey were not designed to be representative of the US population, our findings are consistent with other surveys. Our results do not conflict with national data or suggest a unique set of beliefs among our sample. We believe that our study has added new information to the literature by utilizing rigorous analytic methods including multivariate models such as Lasso regression, which has not been used in previously published studies in this field.

TrackCOVID was one of the few population-based representative surveys of SARS-CoV-2 incidence and prevalence performed in the US at the time.  An address-based sampling frame was used, with stratification by county and risk to ensure precision of estimates. Response to the nested vaccine survey described in this paper was high, 83%. As such, the vaccine study sample was fairly representative of the parent cohort; differences between those who responded to the survey and those who did not, are described in the Results section.

Reference:

  1. Gadoth A, Halbrook M, Martin-Blais R, et al. Assessment of COVID-19 Vaccine Acceptance among Healthcare Workers in Los Angeles. Public and Global Health; 2020. doi:10.1101/2020.11.18.20234468
  2. Unroe, K. T., Evans, R., Weaver, L., Rusyniak, D., & Blackburn, J. (2021). Willingness of Long‐Term Care Staff to Receive a COVID‐19 Vaccine: A Single State Survey. Journal of the American Geriatrics Society, 69(3), 593-599.
  3. Thompson, H. S., Manning, M., Mitchell, J., Kim, S., Harper, F. W., Cresswell, S., ... & Marks, B. (2021). Factors Associated With Racial/Ethnic Group–Based Medical Mistrust and Perspectives on COVID-19 Vaccine Trial Participation and Vaccine Uptake in the US. JAMA Network Open, 4(5), e2111629-e2111629.

Reviewer 2 Report

The authors followed most of the suggestions proposed and the paper has improved compared to the previous version. Although I am moderately satisfied, I believe it necessary to reconsider what was requested in the previous round with respect to the fear of COVID-19 and the restrictive measures with the related citations.

Given the high number of articles that analyze every aspect of the pandemic, it is necessary to differentiate as much as possible from articles already present in the literature.

Author Response

Response:

We appreciate the reviewer’s insightful suggestions.

We agree with the reviewer that it is important to include discussions on the impact of the fear of COVID-19 and the restrictive measures on vaccine willingness during the early period of pandemic. We incorporated the following revisions in the discussion section.

“[Line 276]: Fear of COVID-19 infection and the potential economic, physical and mental health consequences associated with restrictive measures, such as quarantine, lockdown, stay-at-home orders, due to continued transmission, could also prompt people to get vaccinated, although our study did not evaluate these factors31-38. “

As we mentioned in the discussion section, our study provides additional information to previous studies, including by evaluating mediating variables and novel statistical regression analyses.

“[Line 268] This is one of the few published studies3, 21, 26, 29 to test the independent contribution of a broad range of beliefs as predictors of COVID-19 vaccination intentions. In addition, the study applied both multivariate regression techniques and statistical learning methods to identify potential mediators of racial-ethnic differences as well as predictors in COVID-19 vaccination intention.”

We expanded the literature review as suggested and incorporated additional citations as below.

Reference:

  1. Bendau, A., Plag, J., Petzold, M. B., & Ströhle, A. (2021). COVID-19 vaccine hesitancy and related fears and anxiety. International immunopharmacology, 97, 107724.
  2. Killgore, W. D., Cloonan, S. A., Taylor, E. C., & Dailey, N. S. (2021). The COVID-19 Vaccine Is Here—Now Who Is Willing to Get It?. Vaccines, 9(4), 339.
  3. Wake, A. D. (2021). The willingness to receive COVID-19 vaccine and its associated factors:“vaccination refusal could prolong the war of this pandemic”–a systematic review. Risk management and healthcare policy, 14, 2609.
  4. Napoli, P. E., Nioi, M., & Fossarello, M. (2021). The “quarantine dry eye”: the lockdown for coronavirus disease 2019 and its implications for ocular surface health. Risk management and healthcare policy, 14, 1629.
  5. Pfefferbaum, B., & North, C. S. (2020). Mental health and the Covid-19 pandemic. New England Journal of Medicine, 383(6), 510-512.
  6. Kumar, A., & Nayar, K. R. (2021). COVID 19 and its mental health consequences.
  7. Andrade, C. (2020). COVID-19 and lockdown: Delayed effects on health. Indian Journal of Psychiatry, 62(3), 247.
  8. Nioi, M., Napoli, P. E., Finco, G., Demontis, R., Fossarello, M., & d'Aloja, E. (2021). Fear of the COVID-19 and medical liability. Insights from a series of 130 consecutives medico-legal claims evaluated in a single institution during SARS-CoV-2-related pandemic.